# Development of a Short-Form Hwa-Byung Symptom Scale Using Machine Learning Approaches

**DOI:** 10.3390/diagnostics14212419

**Published:** 2024-10-30

**Authors:** Chan-Young Kwon, Boram Lee, Sung-Hee Kim, Seok Chan Jeong, Jong-Woo Kim

**Affiliations:** 1Department of Oriental Neuropsychiatry, College of Korean Medicine, Dong-Eui University, Busan 47227, Republic of Korea; 2KM Science Research Division, Korea Institute of Oriental Medicine, Daejeon 34054, Republic of Korea; qhfka9357@kiom.re.kr; 3Department of Industrial ICT Engineering, Dong-Eui University, Busan 47340, Republic of Korea; sh.kim@deu.ac.kr; 4Grand ICT Research Center, Department of e-Business and Graduate School of Artificial Intelligence, Dong-Eui University, Busan 47340, Republic of Korea; scjeong@deu.ac.kr; 5Department of Korean Neuropsychiatry, Kyung-Hee University Hospital at Gangdong, Seoul 05278, Republic of Korea; aromaqi@naver.com

**Keywords:** anger syndrome, hwa-byung, Republic of Korea, Korean medicine, machine learning

## Abstract

**Background/Objectives**: Hwa-byung (HB), also known as “anger syndrome” or “fire illness”, is a culture-bound syndrome primarily observed among Koreans. This study aims to develop a short-form version of the HB symptom scale using machine learning approaches. Methods: Utilizing exploratory factor analysis (EFA) and various machine learning techniques (i.e., XGBoost, Logistic Regression, Random Forest, Support Vector Machine, Decision Tree, and Multi-Layer Perceptron), we sought to create an efficient HB assessment tool. A survey was conducted on 500 Korean adults using the original 15-item HB symptom scale. **Results**: The EFA revealed two distinct factors: psychological symptoms and somatic manifestations of HB. Statistical testing showed no significant differences between using different numbers of items per factor (ANOVA: F = 0.8593, *p* = 0.5051), supporting a minimalist approach with one item per factor. The resulting two-item short-form scale (Q3 and Q10) demonstrated high predictive power for the presence of HB. Multiple machine learning models achieved a consistent accuracy (90.00% for most models) with high discriminative ability (AUC = 0.9436–0.9579), with the Multi-Layer Perceptron showing the highest performance (AUC = 0.9579). The models showed balanced performance in identifying both HB and non-HB cases, with precision and recall values consistently around 0.90. **Conclusions**: The findings of this study highlighted the effectiveness of integrating EFA and artificial intelligence via machine learning in developing practical assessment tools. This study contributes to advancing methodological approaches for scale development and offers a model for creating efficient assessments of Korean medicine.

## 1. Introduction

Hwa-byung (HB), also known as “anger syndrome” or “fire illness”, is a culture-bound syndrome primarily observed among Koreans [1,2]. It is characterized by somatic, emotional, and cognitive symptoms, often associated with chronic stress and suppressed anger [1]. Common symptoms include palpitations, chest tightness, headaches, and feelings of heat or anger [1]. Korean Confucian culture considers suppression of emotions such as anger to be a virtue, and since this is the pathogenesis of HB, this disease has been considered a culture-bound syndrome or Korean anger syndrome [2]. Patients with HB experience chronic anger through complex cognitive processes involving blame and a feeling of loss of control due to emotional dysregulation and somatic symptoms [3]. According to a community survey, the prevalence of HB is 5.4% [4], and risk factors include female sex, older age, vulnerable traits, and psychosocial stress such as martial conflicts, financial loss, or poverty [2].

Since this mental disorder is closely related to the mental health of Koreans, tools or related documents for diagnosing, evaluating, and treating it, including the HB clinical practice guidelines, have been developed [5,6,7,8]. In particular, HB scales, consisting of the HB trait scale and HB symptom scale, have been developed in previous studies. The HB symptom scale developed by Kwon et al. [5] has been widely used in clinical practice and clinical studies [9,10], as a cut-off score (i.e., 30 points) for HB screening has been developed. This tool consists of 15 items and allows self-assessment of typical psychological and somatic symptoms of HB [5]. This tool has been used in research to compare HB with other mental disorders, including major depressive disorder and generalized anxiety disorder [11]. Some somatic symptoms that appear along with psychological symptoms are the unique characteristic of HB that distinguish it from depressive disorders, including epigastric mass, palpitation, and headache [8]. Some studies pointed to a genetic link between these somatic symptoms of HB [12,13]. Recently, Kim et al. attempted to develop a revised HB scale, but this scale has not yet been officially published [14,15]. They describe that their attempt is not to develop a short-form HB scale, but rather a revision to reflect the current clinical environment [15].

Although these scales, including the HB symptom scale [5], provide detailed insights, they are often lengthy and time consuming, posing a challenge in busy clinical settings and epidemiological studies. In addition, the National Mental Health Survey of Korea conducted every five years in South Korea includes major mental disorders such as depressive and anxiety disorders, but does not include HB or anger-related mental issues [16]. This is probably related to the absence of an optimized short scale that suggests the presence of HB. These limitations make nationwide surveillance and longitudinal tracking of changes in this culture-bound syndrome difficult. Accordingly, there is an evident need for a shorter yet equally reliable assessment tool to facilitate quicker screening and efficient management of HB.

Identification of timely mental health problems of publics is an important part of establishing and implementing national mental health policies [17]. Given the recent increase in anger-related problems in South Korea [18,19,20], developing a tool to quickly and accurately screen for HB, an anger syndrome that reflects Korean cultural characteristics, is an urgent and relevant challenge. In addition, the development of such a tool could enable HB to be reflected in overall mental health assessment tools as well as investigating HB in large-scale epidemiological studies. Another potential benefit is that, as the development of mobile health drives mental health services [21], the development of a short-form scale of HB could have a larger and broader impact on the mental health of the public, as it is easily delivered via mobile technology.

Advancements in statistical and machine learning techniques offer new opportunities for scale development. Exploratory factor analysis (EFA) helps identify the underlying structure of a set of variables, making it possible to pinpoint the most representative items [22]. Furthermore, machine learning algorithms facilitate the creation of predictive models [23] that can estimate scores of comprehensive scales based on a few selected items, maintaining the accuracy and reliability of the assessment [24,25]. For example, a study by Wall et al. reported promising results using machine learning to reduce 93 questions to 7 questions in an autism diagnostic interview [26]. Despite advancements in machine learning, the varying performance of different models based on dataset characteristics and specific tasks highlights the need for thorough comparisons to identify effective approaches, improve model selection, and advance methodologies for practical applications.

Therefore, this study aims to develop a short-form version of the HB symptom scale by utilizing EFA to identify key items and several machine learning models to predict the presence of HB based on this short-form scale. Thus, we strived to create a practical and efficient tool for HB assessment that can be easily implemented in various settings. This article covers the study’s participants, measures, and methods, followed by the results of factor analysis and model performance. It then discusses the findings, interpretations, and limitations, and concludes with implications and future research directions.

## 2. Materials and Methods

### 2.1. Participants

An anonymous survey of 500 Korean adults in the general population was conducted by Macromill Embrain (Embrain Co., Ltd., Seoul, Republic of Korea). The survey was conducted between 7 June and 12, 2024. The inclusion criteria were being 19–44 years of age (i.e., the MZ generation born between 1980 and 2005) and having sufficient literacy to understand and complete the survey. The participants were recruited from the general population, and there were no restrictions on their gender, race, current medical history, or past medical history. The reason for targeting the MZ generation is that this generation is the main consumer generation in South Korea today, and it is suspected that this generation will be most affected by HB due to its sensitivity to unfairness and injustice [18,27]. The original purpose of the survey was to investigate the prevalence of HB and related factors in the Korean MZ generation, and the current study was conducted as a secondary analysis of that survey.

### 2.2. Measures

The original HB symptom scale consists of 15 items designed to capture the multidimensional nature of HB, including psychological and somatic symptoms [5]. Each item is rated on a 5-point Likert scale (i.e., not at all, 0 points; not really, 1 point; moderately, 2 points; quite a bit, 3 points; and completely, 4 points), with higher scores indicating greater symptom severity [5]. The scale has been previously validated and widely used in clinical settings and clinical research [5,9]. Importantly, the cut-off score for this scale, which suggests the presence of HB, is 30 points. The 15-item HB symptom scale, which consists of psychological and somatic symptoms, is presented in Table 1. Each item represents a specific aspect of HB symptomatology, with items 1–5, 8, and 15 primarily capturing psychological symptoms, while items 7, 9–13 focus on somatic manifestations. Items 6 and 14 have characteristics of both psychological and somatic symptoms.

### 2.3. Statistical Analysis and Machine Learning Algorithms

#### 2.3.1. Data Preprocessing

The dataset was loaded from a CSV file containing the responses to the 15-item HB symptom scale. Initial data exploration involved checking for data completeness, the distribution of responses, and potential outliers. Missing values were handled by substituting them with the mean value of each respective item. This approach ensured that no data points were lost while maintaining the overall distribution of the data.

#### 2.3.2. Item Grouping with EFA

EFA was conducted to identify the underlying latent factors in the HB symptom scale. The optimal number of factors was determined using parallel analysis [28], which compared the eigenvalues of the observed data with those of randomly generated data. This method is preferred over traditional approaches like the Kaiser criterion or a scree plot as it provides a more robust and objective determination of factor numbers [28].

Varimax rotation was applied to improve the interpretability of the factor loadings. Factor analysis was performed using the Python factor analyzer package. The optimal number of factors was determined using Kaiser criterion (eigenvalues > 1) [29]. Then, to determine the optimal number of items to include for each factor (*k*), we employed a cross-validation approach using the XGBoost classifier. XGBoost is a popular gradient boosting algorithm that uses an ensemble of decision trees to create a robust predictive model, offering high performance and computational efficiency through techniques such as regularization and parallel processing [24]. Values of *k* ranging from 1 to 5 were tested using 5-fold cross-validation to predict the original HB diagnosis (total score ≥ 30). For each *k* value, the top *k* items with the highest loadings were selected from each factor, and the XGBoost model’s predictive performance was evaluated using cross-validation scores. Statistical comparisons between cross-validation scores for different *k* values were performed using one-way Analysis of Variance (ANOVA) and paired *t*-tests to determine if increasing *k* resulted in statistically significant improvements in predictive performance. If increasing the *k* value did not lead to a statistically significant improvement in predictive performance, we chose the minimum *k* value. This iterative process allowed us to balance model simplicity and predictive power, with the highest cross-validation score determining the optimal *k* value.

#### 2.3.3. Model Development and Evaluation

Multiple machine learning approaches were adopted to predict the presence of HB using the short-form scale. Six different algorithms were implemented: XGBoost, Logistic Regression (LR), Random Forest (RF), Support Vector Machine (SVM), Decision Tree (DT), and Multi-Layer Perceptron (MLP). Each algorithm was selected for its specific strengths in classification tasks. For each model, hyperparameter tuning was performed using GridSearchCV with 5-fold cross-validation. The following hyperparameters were optimized: (a) XGBoost: n_estimators [100, 200], max_depth [3, 5, 7], and learning_rate [0.01, 0.1, 0.2]; (b) LR: C [0.1, 1, 10]; (c) RF: n_estimators [100, 200] and max_depth [3, 5, 7]; (d) SVM: kernel [‘linear’, ‘rbf’]; (e) DT: max_depth [3, 5, 7] and min_samples_split [2, 5, 10]; and (f) MLP: hidden_layer_sizes [(50,), (100,), (50, 50)] and activation [‘tanh’, ‘relu’], with fixed learning_rate_init = 0.01. The dataset was divided into training (70%) and testing (30%) sets [30] using stratified sampling to maintain the proportion of HB cases in both sets. The stratification ensured that the class distribution was preserved in both training and testing sets, which is crucial for imbalanced datasets.

#### 2.3.4. Performance Measures

The performance of each model was evaluated using several standard metrics. For a binary classification problem with True Positives (TPs), True Negatives (TNs), False Positives (FPs), and False Negatives (FNs), the following measures were calculated: (a) accuracy = (TP + TN)/(TP + TN + FP + FN); (b) precision (positive predictive value) = TP/(TP + FP); (c) recall (sensitivity) = TP/(TP + FN); and (d) F1-score = 2 × (precision × recall)/(precision + recall). Additionally, the Area Under the Receiver Operating Characteristic Curve (AUC-ROC) was calculated to assess the models’ discriminative ability across different classification thresholds. The ROC curve is created by plotting the true positive rate (TPR = TP/(TP + FN)) against the false positive rate (FPR = FP/(FP + TN)) at various threshold settings. Several visualizations were created to aid in the interpretation of the results: (a) ROC curves comparing the performance of all models and (b) learning curves showing training and validation scores across different training set sizes.

### 2.4. Software and Tools

The analysis was conducted using Python version 3.12.2. The key libraries included pandas (ver. 2.2.3) for data manipulation and analysis, scikit-learn (ver. 1.5.2) for implementing multiple machine learning algorithms (i.e., LR, RF, SVM, DT, and MLP) and preprocessing tools, XGBoost (ver. 2.1.1) for the gradient boosting model, factor_analyzer (ver. 0.5.1) for EFA, seaborn (ver. 0.13.2) and matplotlib (ver. 3.9.2) for data visualization, GridSearchCV (from scikit-learn) for hyperparameter optimization, and numpy (ver. 2.1.2) for numerical computations. The code was structured in a modular fashion with separate functions for data preprocessing, factor analysis, model training, and evaluation to facilitate code maintenance and replication of results.

### 2.5. Ethical Considerations

The research protocol was reviewed and approved by the Institutional Review Board of the Dong-eui University Korean Medicine Hospital (DH-2024-09, approved on 14 August 2024).

## 3. Results

### 3.1. Data Description

The dataset comprised responses from 500 participants, each of whom completed a 15-item HB symptom scale. The demographics of the participants were as follows: 47.4% female (237/500), average age of 34.74 (±7.43) years, and average total score of the HB symptom scale of 26.67 (±12.14). Among the participants, 37% (195/500) had a total score of 30 or higher on the HB symptom scale, implying the presence of HB (Figure 1).

### 3.2. HB Symptom Scale Items Grouped by Factor Composition

A parallel analysis suggested an optimal number of two factors for the HB symptom scale. The factor loadings after the varimax rotation revealed the following groupings. Factor 1 showed high loadings (>0.6) for items Q1, Q2, Q3, Q4, Q5, and Q15. This factor generally represents psychological symptoms of HB. Factor 2 had high loadings (>0.6) for items Q6, Q7, Q9, Q10, and Q11, which mainly represent somatic manifestations of HB (Table 2). Cross-validation analysis was performed to determine the optimal number of items (*k*) per factor, and the optimal *k* value was determined to be 5. However, no statistically significant difference was found in the one-way ANOVA and paired *t*-tests according to the change in *k* value. Following the principle of parsimony, *k* = 1 was selected as the optimal value, resulting in two items: Q3 (psychological factor) and Q10 (somatic factor) (Table 3).

### 3.3. Prediction of the Presence of HB

All six machine learning models demonstrated robust performance with the two-item short-form scale, achieving an identical accuracy of 90.00% for five models, XGBoost, RF, SVM, DT, and MLP, while LR exhibited a slightly lower accuracy of 89.33%. The specific performance metrics revealed that XGBoost achieved an AUC of 0.9549, with a precision and recall of 0.89 and 0.94 for non-HB cases, and 0.91 and 0.84 for HB cases, respectively. RF achieved an AUC of 0.9567, with identical precision and recall metrics to XGBoost. SVM produced an AUC of 0.9436, with performance metrics identical to RF. The DT model also showed an AUC of 0.9453, similar to the performance metrics of RF. The MLP model achieved an AUC of 0.9579, maintaining identical performance metrics. Lastly, LR yielded an AUC of 0.9528, with a precision and recall of 0.87 and 0.97 for non-HB cases, and 0.94 and 0.79 for HB cases. The nearly identical performance across various models suggests that the two-item short-form scale provides robust and stable predictions of HB presence, regardless of the machine learning algorithm employed (Table 4; Figure 2).

## 4. Discussion

### 4.1. Findings of This Study

This study successfully developed a short-form version of the HB symptom scale (Q3 and Q10) using EFA and machine learning techniques. We identified two distinct factors underlying the HB symptom scale utilizing EFA: (a) psychological symptoms and (b) somatic manifestations. Through statistical testing (ANOVA: F = 0.8593, *p* = 0.5051) and paired *t*-tests, we found no significant differences between using one item per factor versus using more items, supporting the use of a single representative item per factor (*k* = 1). Among the various models evaluated, most models (XGBoost, RF, SVM, DT, and MLP) achieved identical accuracies of 90.00%, with only LR showing slightly lower accuracy at 89.33%. The MLP model demonstrated the highest AUC (0.9579), followed closely by RF (AUC = 0.9567) and XGBoost (AUC = 0.9549). All models showed balanced performance in identifying both HB and non-HB cases, with precision and recall values consistently around 0.90.

### 4.2. Interpretation of the Findings

The prevalence of HB among the participants in this survey was 37%, which is significantly higher than the 5.4% found in previously reported epidemiological studies [4]. These findings are thought to be because the participants targeted the MZ generation, which is considered to be sensitive to unfairness and injustice and thus vulnerable to HB. As evidence, a recent study found that 47.3% of the Korean general public reported “persistent” or “increased” embitterment, related to suppressed anger, the core pathology of HB [19]. The difference between HB and embitterment is that HB includes somatic symptoms such as palpitations, chest tightness, headaches, and feelings of heat in addition to psychological symptoms [8]. The discrepancy with previously reported prevalence rates of HB [4] may reflect broader mental health changes in Korean society. The previous study was conducted about 15 years ago [4], and factors such as the coronavirus pandemic may have contributed to worse mental health outcomes [31,32], including an increased prevalence of HB.

The two-factor structure identified in our study provides a deeper understanding of HB symptomatology. Specifically, the identification of two factors in EFA—psychological symptoms and somatic manifestations—supports the multidimensional nature of HB. The psychological factor, represented by item Q3 (“I feel that my life is sorrowful”), captures the core emotional distress associated with HB. Represented by item Q10 (“I often feel a heat buildup in my chest”), the somatic factor reflects the unique physical manifestations of HB, distinguishing it from purely psychological conditions, such as embitterment and intermittent explosive disorder. This factor structure aligns with the existing literature that describes HB as encompassing both psychological and physical components [1,2,9]. By focusing on these two core aspects, this study reaffirms the validity of the original scale while offering a more practical assessment tool.

The use of cross-validation and statistical testing to determine the optimal number of items (*k* = 1) per factor was a crucial advancement. Our analysis showed no significant differences between using one item per factor versus using more items, supporting our minimalist approach. Selecting the most representative item for each factor ensured that the short-form scale captured both the psychological and somatic symptoms while maintaining simplicity. This approach simplified the scale without compromising its predictive ability, as demonstrated by the high accuracy (90.00%) achieved consistently across multiple machine learning models. The shortened two-item scale’s robust performance demonstrates that a minimalistic approach can be both effective and efficient. By developing a shorter version of the HB symptom scale, this study addressed the need for more practical and less time-consuming assessment tools. This can significantly benefit clinical practice by allowing faster screening and reducing the burden on both patients and healthcare providers.

The study design incorporated measures to mitigate the risk of overfitting, a common concern in machine learning approaches [33]. By partitioning the dataset into separate training (70%) and testing (30%) subsets [30], we ensured that the model’s performance was evaluated on data it had not encountered during the training phase. This approach allowed for a more robust assessment of the model’s generalizability. The use of stratified sampling in this division process maintained the proportion of HB cases across both subsets, further enhancing the reliability of our results. Additionally, the implementation of cross-validation during the hyperparameter tuning phase provided an extra layer of protection against overfitting. These methodological choices collectively strengthen the validity of our findings and the potential applicability of the short-form HB symptom scale in real-world settings.

The application of multiple machine learning approaches demonstrates the robustness of our short-form scale. The consistent performance across different models (XGBoost, RF, SVM, DT, and MLP all achieving 90.00% accuracy) underscores the reliability of using these two items for HB prediction. Particularly noteworthy was the high AUC values across all models (ranging from 0.9436 to 0.9579), indicating excellent discriminative ability. This consistency across different algorithms not only confirms the feasibility of using machine learning for scale reduction but also illustrates how these techniques can improve practical usability in clinical and research settings. The combination of EFA and machine learning not only advances methodological approaches in scale development but also offers a model for future research in other culturally specific syndromes or psychological assessments.

The performance of our machine learning models can be compared with similar studies that used machine learning for scale reduction in psychiatric assessments. For instance, Wall et al. [26] achieved 99.9% accuracy in reducing the 93-item autism diagnostic interview to 7 items using machine learning. More recently, Lin et al. [34] reported an AUC of 0.81 when developing a five-item short form of the children’s depression inventory using various machine learning approaches. Similarly, Jo et al. [25] achieved an accuracy of 90% in developing a shortened version of the Dysfunctional Beliefs and Attitudes about Sleep scale. In comparison, our study achieved higher performance metrics (90.00% accuracy and AUC values of 0.9436–0.9579) while reducing the scale to just two items. This superior performance might be attributed to several factors: (a) the clear two-factor structure of the original HB scale, (b) the use of multiple advanced machine learning algorithms with optimized hyperparameters, and (c) the distinct characteristics of HB symptoms that make them more amenable to accurate classification. These comparisons suggest that our approach not only achieves state-of-the-art performance but also demonstrates the possibility of developing short-form screening tools without significant loss of diagnostic accuracy.

In summary, our findings underscore the effectiveness of integrating EFA and machine learning to create a practical and reliable HB assessment tool. This study’s approach to factor analysis, item selection, and predictive modeling sets a precedent for developing efficient scales in various fields of psychology and medicine.

### 4.3. Limitations

Despite these promising findings, this study had several limitations. First, the study sample consisted of Korean adults aged 19–44 years. These results may not be generalizable to other age groups or cultural contexts. Future studies should include more diverse populations to validate our findings across different demographics. Second, although our statistical analysis supported the use of just two items with no significant performance differences from using more items (*p* > 0.05), future research should continue to explore the stability of this short form across different populations and settings. Third, this study relied on self-reported data, which can be subject to recall bias and social desirability bias [35]. Incorporating clinician assessments in future research could provide a more comprehensive evaluation of HB symptoms. Fourth, this study was conducted on 500 samples, 70% of which were used for model development and the remaining 30% for model validation. Future studies with larger sample sizes could potentially improve the performance of the developed model. Additionally, longitudinal studies could help assess the stability of HB symptoms over time and the predictive validity of the short-form scale in capturing changes in symptom severity.

## 5. Conclusions

In conclusion, this study successfully developed a short-form version of the HB symptom scale using both EFA and machine learning techniques. This short-form scale, consisting of only two items—Q3 (psychological symptoms) and Q10 (somatic symptoms)—retained the diagnostic integrity of the original 15-item scale while significantly enhancing efficiency in clinical and research settings. Statistical testing supported this minimalist approach, showing no significant differences between using two items versus more items. The consistent performance across multiple machine learning models (90.00% accuracy for most models) and high AUC values (0.9436–0.9579) demonstrated the robustness of this short-form scale. The development of this short-form scale holds great promise for future applications. By reducing the time and effort needed for HB screening, this tool can be readily used in large-scale epidemiological studies and busy clinical settings, where the original 15-item scale may not be practical. Future research should focus on validating the short-form HB scale across different age groups, populations, and cultural contexts to confirm its broader applicability and reliability.

## Figures and Tables

**Figure 1 diagnostics-14-02419-f001:**
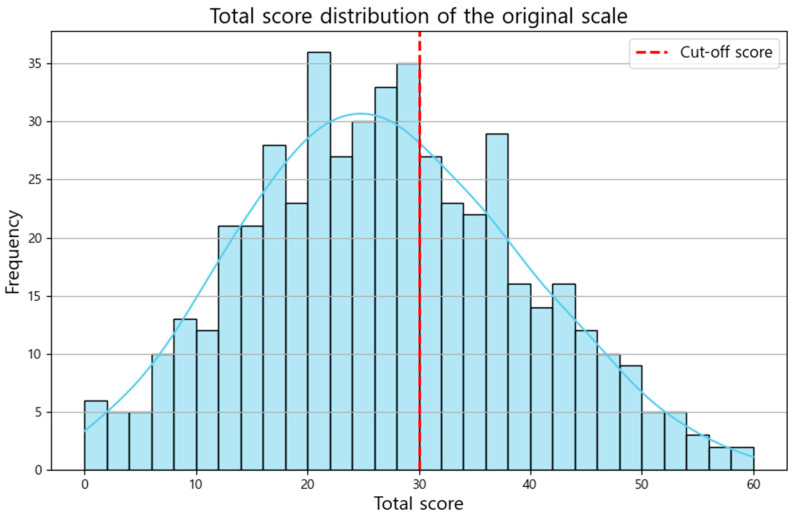
Distribution of HB symptom scores of the participants. Abbreviations. HB, hwa-byung.

**Figure 2 diagnostics-14-02419-f002:**
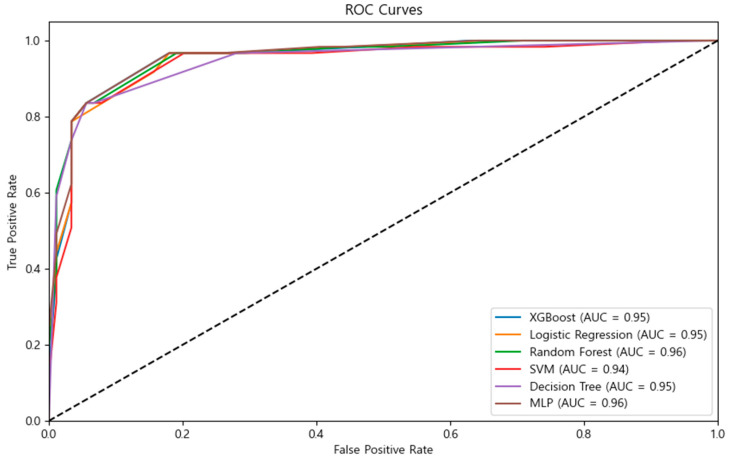
ROC curve of true positive rate and false positive rate. Abbreviations. AUC, area under the curve; MLP, Multi-Layer Perceptron; ROC, receiver operating characteristic; SVM, Support Vector Machine.

**Table 1 diagnostics-14-02419-t001:** The 15 items of the HB symptom scale and their characteristics.

Questions		Psychological	Somatic
1	My life is rather unhappy.	O	
2	There are times when I feel deep regret or resentment.	O	
3	I feel that my life is sorrowful.	O	
4	I feel sorrowful.	O	
5	I feel wronged.	O	
6 *	My nerves are so fragile that I can’t control my emotions.	O	O
7	My hands and feet tremble, and I feel restless.		O
8	I often feel disappointed in myself.	O	
9	My face often flushes with heat.		O
10	I often feel a heat buildup in my chest.		O
11	I often feel something rising from below (legs or abdomen) to above (chest).		O
12	When I get angry, my hands feel numb or tremble.		O
13	I have indigestion and often feel bloated.		O
14 *	I am very exhausted.	O	O
15	I feel that the world is unfair.	O	

* These items have the characteristics of both psychological and somatic symptoms.

**Table 2 diagnostics-14-02419-t002:** Factor loadings of HB symptom scale items.

Questions		Factor 1	Factor 2
1	My life is rather unhappy.	**0.70**	0.38
2	There are times when I feel deep regret or resentment.	**0.77**	0.27
3	I feel that my life is sorrowful.	**0.83**	0.36
4	I feel sorrowful.	**0.82**	0.37
5	I feel wronged.	**0.70**	0.44
6	My nerves are so fragile that I can’t control my emotions.	0.45	**0.63**
7	My hands and feet tremble, and I feel restless.	0.34	**0.70**
8	I often feel disappointed in myself.	0.58	0.43
9	My face often flushes with heat.	0.31	**0.66**
10	I often feel a heat buildup in my chest.	0.36	**0.80**
11	I often feel something rising from below (legs or abdomen) to above (chest).	0.30	**0.77**
12	When I get angry, my hands feel numb or tremble.	0.27	0.59
13	I have indigestion and often feel bloated.	0.29	0.54
14	I am very exhausted.	0.42	0.35
15	I feel that the world is unfair.	**0.60**	0.31

Note. Bold values indicate factor loadings > 0.60.

**Table 3 diagnostics-14-02419-t003:** Cross-validation scores according to *k* values.

*k* Value	Number of Selected Items	Cross-Validation Score (±SD)	t-Value (*p*-Value)	ANOVA (*p*-Value)
1	2	0.8920 (±0.0248)	NA	F = 0.8593 (0.5051)
2	4	0.9160 (±0.0307)	*k* = 1 vs. *k* = 2: t = −2.7530 (0.0512)	
3	6	0.9140 (±0.0206)	*k* = 1 vs. *k* = 3: t = −1.5039 (0.2070)	
4	8	0.9100 (±0.0303)	*k* = 1 vs. *k* = 4: t = −0.9705 (0.3868)	
5	10	0.9240 (±0.0162)	*k* = 1 vs. *k* = 5: t = −1.9332 (0.1254)	

Abbreviations. ANOVA, Analysis of Variance; HB, hwa-byung; NA, not applicable; SD, standard deviation.

**Table 4 diagnostics-14-02419-t004:** Performance of different machine learning models in predicting HB presence using the short-form scale.

Model	Class	Precision	Recall	F1-Score	Support
XGBoost	No HB	0.89	0.94	0.92	89
	HB	0.91	0.84	0.87	61
	Accuracy			0.90	150
	Macro avg	0.90	0.89	0.89	150
	Weighted avg	0.90	0.90	0.90	150
LR	No HB	0.87	0.97	0.91	89
	HB	0.94	0.79	0.86	61
	Accuracy			0.89	150
	Macro avg	0.90	0.88	0.89	150
	Weighted avg	0.90	0.89	0.89	150
RF	No HB	0.89	0.94	0.92	89
	HB	0.91	0.84	0.87	61
	Accuracy			0.90	150
	Macro avg	0.90	0.89	0.89	150
	Weighted avg	0.90	0.90	0.90	150
SVM	No HB	0.89	0.94	0.92	89
	HB	0.91	0.84	0.87	61
	Accuracy			0.90	150
	Macro avg	0.90	0.89	0.89	150
	Weighted avg	0.90	0.90	0.90	150
DT	No HB	0.89	0.94	0.92	89
	HB	0.91	0.84	0.87	61
	Accuracy			0.90	150
	Macro avg	0.90	0.89	0.89	150
	Weighted avg	0.90	0.90	0.90	150
MLP	No HB	0.89	0.94	0.92	89
	HB	0.91	0.84	0.87	61
	Accuracy			0.90	150
	Macro avg	0.90	0.89	0.89	150
	Weighted avg	0.90	0.90	0.90	150

Abbreviations. DT, Decision Tree; HB, hwa-byung; LR, Logistic Regression; MLP, Multi-Layer Perceptron; RF, Random Forest; SVM, Support Vector Machine.

## Data Availability

The data that support the findings of this study are available from the corresponding author upon reasonable request.

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
