# Peer review of "Development of a Short-Form Hwa-Byung Symptom Scale Using Machine Learning Approaches"

_diagnostics, 2024, doi:10.3390/diagnostics14212419_

Round 1

Reviewer 1 Report

Comments and Suggestions for Authors

The article entitled “Development of a short-form hwa-byung symptom scale using a machine learning approach” is well-written and, from my point of view, would be of interest for the readers of Diagnostics.

The study developed a short version of the Hwa-byung Symptom Scale using exploratory factor analysis and the XGBoost model. Two factors were identified: psychological symptoms and somatic manifestations. The two-item scale showed high accuracy in detecting the syndrome. The need for more efficient assessment tools for the diagnosis and management of Hwa-byung is highlighted.

In spite of this, and before its publication, I would suggest authors to perform the following changes:

In Figure 1 entitled “HB symptom scale items groups by factor composition. Note. The numbers in the cells 182 represent the loading values for each item” there is certain Information that does not require of colours and would be included in a simple table.

I really appreciate the discussion and limitations sections of the manuscript but the section that corresponds to conclusions should be more detailed.

Author Response

  • Comments from Reviewer 1:

Overall comment:

The article entitled “Development of a short-form hwa-byung symptom scale using a machine learning approach” is well-written and, from my point of view, would be of interest for the readers of Diagnostics.

The study developed a short version of the Hwa-byung Symptom Scale using exploratory factor analysis and the XGBoost model. Two factors were identified: psychological symptoms and somatic manifestations. The two-item scale showed high accuracy in detecting the syndrome. The need for more efficient assessment tools for the diagnosis and management of Hwa-byung is highlighted.

In spite of this, and before its publication, I would suggest authors to perform the following changes:

Response:              

Thank you for taking the time to review this manuscript. We have revised this manuscript as faithfully as possible, taking into account your comments.

Comment 1:

In Figure 1 entitled “HB symptom scale items groups by factor composition. Note. The numbers in the cells 182 represent the loading values for each item” there is certain Information that does not require of colours and would be included in a simple table.

Response 1:           

We appreciate the reviewer’s suggestion to improve Figure 1. We have transformed the figure into a clear and readable table format, removing unnecessary colors, which ensures that the information is conveyed concisely.

(Please see Table 2, in page 6)

Comment 2:

I really appreciate the discussion and limitations sections of the manuscript but the section that corresponds to conclusions should be more detailed.

Response 2:           

We thank the reviewer for the feedback on the conclusions. We have expanded the conclusions to provide a more comprehensive discussion of the study's implications, practical applications, and future directions.

“In conclusion, this study successfully developed a short-form version of the HB symptom scale using both EFA and machine learning techniques. This short-form scale, consisting of only two items—Q3 (psychological symptoms) and Q10 (somatic symptoms)—retained the diagnostic integrity of the original 15-item scale while significantly enhancing efficiency in clinical and research settings. Statistical testing supported this minimalist approach, showing no significant differences between using two items versus more items. The consistent performance across multiple machine learning models (90.00% accuracy for most models) and high AUC values (0.9436-0.9579) demonstrated the robustness of this short-form scale. The development of this short-form scale holds great promise for future applications. By reducing the time and effort needed for HB screening, this tool can be readily used in large-scale epidemiological studies and busy clinical settings, where the original 15-item scale may not be practical. Future research should focus on validating the short-form HB scale across different age groups, populations, and cultural contexts to confirm its broader applicability and reliability.”

(Please see red words, in page 10)

Reviewer 2 Report

Comments and Suggestions for Authors

The following points need to be addressed in the revision

Abstract

  1. The abstract is very well written. However, some suggestions may further improve it.
  2. The advancement / novelty in the methodology still needs to be highlighted. However, the study reveals the excellent utility of AI via ML in diagnosing the HB scale.
  3. The public availability of dataset for the reproducibility of results may be mentioned.
  4. A comparison with SOTA techniques should be made in the abstract.
  5. Future study recommendations should be removed from the abstract to prove its usefulness. The text related to future work may be transferred to a subsection at the end of the Results section (before Conclusions). The text to be adjusted is “Further research is required to validate the short-form scale across diverse populations and contexts.”

Introduction

  1. The main innovation and contribution of this research should be clarified in the introduction.
  2. The literature needs to be thoroughly surveyed on the HB symptom scale. The references are insufficient. Add more relevant references to show that the task was thoroughly gone through before proceeding.
  3. The organization of the article may be added in this section.

Materials and Methods

  1. An overview of the XGBoost classifier may be added so that the audience does not need to search elsewhere for it.
  2. Performance measures used in the study may be added as a subsection with mathematical relationships.
  3. Major concern: Why has only a single classifier been used? You should add other strong machine learning classifiers as well.
  4. Major concern: Why did you not try the deep learning strategies?

Results

  1. The result of a single classifier needs to be added by including some of the other strong classifiers.
  2. The findings based on the results of a single classifier need to be made acceptable.
  3. Remove the text like, “Fifth, while the XGBoost algorithm demonstrated high performance in this study, comparing its effectiveness with other machine learning algorithms or traditional statistical methods could offer additional insights into the most appropriate approach for developing short-form scales.”
  4. Major concern: Comparison with SOTA techniques: Can compare the results with similar studies or some previous findings of similar studies.

Comments on the Quality of English Language

Minor language editing would do the job.

Author Response

  • Comments from Reviewer 2:

Overall comment:

The following points need to be addressed in the revision

Response:              

Thank you for taking the time to review this manuscript. We have revised this manuscript as faithfully as possible, taking into account your comments.

[ABSTRACT]

Comment 1:

The abstract is very well written. However, some suggestions may further improve it.

The advancement / novelty in the methodology still needs to be highlighted. However, the study reveals the excellent utility of AI via ML in diagnosing the HB scale.

Response 1:           

Thank you for this valuable suggestion. We have revised the abstract to better emphasize the novel use of machine learning in developing a short-form hwa-byung symptom scale.

Background/Objectives: Hwa-byung (HB), also known as “anger syndrome” or “fire illness,” is a culture-bound syndrome primarily observed among Koreans. This study aims to develop a short-form version of the HB symptom scale using machine learning approaches. Methods: Utilizing exploratory factor analysis (EFA) and various machine learning techniques (i.e., XGBoost, Logistic Regression, Random Forest, Support Vector Machine, Decision Tree, and Multi-Layer Perceptron), we sought to create an efficient HB assessment tool. A survey was conducted on 500 Korean adults using the original 15-item HB symptom scale. Results: The EFA revealed two distinct factors: psychological symptoms and somatic manifestations of HB. Statistical testing showed no significant differences between using different numbers of items per factor (ANOVA: F=0.8593, p=0.5051), supporting a minimalist approach with one item per factor. The resulting two-item short-form scale (Q3 and Q10) demonstrated high predictive power for the presence of HB. Multiple machine learning models achieved a consistent accuracy (90.00% for most models) with high discriminative ability (AUC=0.9436-0.9579), with the Multi-Layer Perceptron showing the highest performance (AUC=0.9579). The models showed balanced performance in identifying both HB and non-HB cases, with precision and recall values consistently around 0.90. Conclusions: The findings of this study highlighted the effectiveness of integrating EFA and artificial intelligence via machine learning in developing practical assessment tools. This study contributes to advancing methodological approaches for scale development and offers a model for creating efficient assessments of Korean medicine.”

(Please see red words, in page 1)

Comment 2:

The public availability of dataset for the reproducibility of results may be mentioned.

Response 2:           

Thank you for the comment. The Data Availability Statement is described on page 10.

Data Availability Statement: The data that support the findings of this study are available from the corresponding author upon reasonable request.”

(Please see red words, in page 10)

Comment 3:

A comparison with SOTA techniques should be made in the abstract.

Response 3:           

We appreciate this insightful suggestion. In this revised manuscript, we have taken your comments into account and compared all six models, including deep learning. This has been fully reflected in the abstract and content of the current manuscript.

Background/Objectives: Hwa-byung (HB), also known as “anger syndrome” or “fire illness,” is a culture-bound syndrome primarily observed among Koreans. This study aims to develop a short-form version of the HB symptom scale using machine learning approaches. Methods: Utilizing exploratory factor analysis (EFA) and various machine learning techniques (i.e., XGBoost, Logistic Regression, Random Forest, Support Vector Machine, Decision Tree, and Multi-Layer Perceptron), we sought to create an efficient HB assessment tool. A survey was conducted on 500 Korean adults using the original 15-item HB symptom scale. Results: The EFA revealed two distinct factors: psychological symptoms and somatic manifestations of HB. Statistical testing showed no significant differences between using different numbers of items per factor (ANOVA: F=0.8593, p=0.5051), supporting a minimalist approach with one item per factor. The resulting two-item short-form scale (Q3 and Q10) demonstrated high predictive power for the presence of HB. Multiple machine learning models achieved a consistent accuracy (90.00% for most models) with high discriminative ability (AUC=0.9436-0.9579), with the Multi-Layer Perceptron showing the highest performance (AUC=0.9579). The models showed balanced performance in identifying both HB and non-HB cases, with precision and recall values consistently around 0.90. Conclusions: The findings of this study highlighted the effectiveness of integrating EFA and artificial intelligence via machine learning in developing practical assessment tools. This study contributes to advancing methodological approaches for scale development and offers a model for creating efficient assessments of Korean medicine.”

(Please see red words, in page 1)

“2.3.3. Model Development and Evaluation

Multiple machine learning approaches were adopted to predict the presence of HB using the short-form scale. Six different algorithms were implemented: XGBoost, Logistic Regression (LR), Random Forest (RF), Support Vector Machine (SVM), Decision Tree (DT), and Multi-Layer Perceptron (MLP). Each algorithm was selected for its specific strengths in classification tasks. For each model, hyperparameter tuning was performed using GridSearchCV with 5-fold cross-validation. The following hyperparameters were optimized: (a) XGBoost: n_estimators [100, 200], max_depth [3, 5, 7], learning_rate [0.01, 0.1, 0.2]; (b) LR: C [0.1, 1, 10]; (c) RF: n_estimators [100, 200], max_depth [3, 5, 7]; (d) SVM: kernel ['linear', 'rbf']; (e) DT: max_depth [3, 5, 7], min_samples_split [2, 5, 10]; and (f) MLP: hidden_layer_sizes [(50,), (100,), (50, 50)], activation ['tanh', 'relu'], with fixed learning_rate_init=0.01. The dataset was divided into training (70%) and testing (30%) sets [30] using stratified sampling to maintain the proportion of HB cases in both sets. The stratification ensured that the class distribution was preserved in both training and testing sets, which is crucial for imbalanced datasets.”

(Please see red words, in page 4)

“All six machine learning models demonstrated robust performance with the two-item short-form scale, achieving an identical accuracy of 90.00% for five models: XGBoost, RF, SVM, DT, and MLP, while LR exhibited a slightly lower accuracy of 89.33%. The specific performance metrics revealed that XGBoost achieved an AUC of 0.9549, with precision and recall of 0.89 and 0.94 for non-HB cases, and 0.91 and 0.84 for HB cases, respectively. RF achieved an AUC of 0.9567 with identical precision and recall metrics as XGBoost. SVM produced an AUC of 0.9436 with performance metrics identical to RF. DT model also showed an AUC of 0.9453, similar to the performance metrics of RF. MLP model reached an AUC of 0.9579, maintaining identical performance metrics. Lastly, LR yielded an AUC of 0.9528, with precision and recall of 0.87 and 0.97 for non-HB cases, and 0.94 and 0.79 for HB cases. The nearly identical performance across various models suggests that the two-item short-form scale provides robust and stable predictions of HB presence, regardless of the machine learning algorithm employed (Table 4; Figure 2).”

(Please see red words, in pages 6-7)

Comment 4:

Future study recommendations should be removed from the abstract to prove its usefulness. The text related to future work may be transferred to a subsection at the end of the Results section (before Conclusions). The text to be adjusted is “Further research is required to validate the short-form scale across diverse populations and contexts.”

Response 4:           

Thanks for your comment. The sentence you pointed out has been removed from the abstract.

[INTRODUCTION]

Comment 5:

The main innovation and contribution of this research should be clarified in the introduction.

Response 5:           

We have revised the Introduction to clarify the innovative aspects of this research, emphasizing the use of machine learning to streamline scale development.

“Advancements in statistical and machine learning techniques offer new opportunities for scale development. Exploratory factor analysis (EFA) helps identify the underlying structure of a set of variables, making it possible to pinpoint the most representative items [22]. Furthermore, machine learning algorithms facilitate the creation of predictive models [23] that can estimate scores of comprehensive scales based on a few selected items, maintaining the accuracy and reliability of the assessment [24,25]. For example, a study by Wall et al. reported promising results using machine learning to reduce 93 questions to 7 questions of autism diagnostic interview [26]. Despite advancements in machine learning, the varying performance of different models based on dataset characteristics and specific tasks highlights the need for thorough comparisons to identify effective approaches, improve model selection, and advance methodologies for practical applications.”

(Please see red words, in page 2)

Comment 6:

The literature needs to be thoroughly surveyed on the HB symptom scale. The references are insufficient. Add more relevant references to show that the task was thoroughly gone through before proceeding.

Response 6:           

We have added several relevant references to strengthen the literature review on the HB symptom scale.

“Since this mental disorder is closely related to the mental health of Koreans, tools or related documents for diagnosing, evaluating, and treating it, including the HB clinical practice guidelines, have been developed [5-8]. Especially, HB scales, consists of HB trait scale and HB symptom scale, have been developed in previous studies. The HB symptom scale developed by Kwon et al. [5] has been widely used in clinical practice and clinical studies [9,10], as a cut-off score (i.e., 30 points) for HB screening has been developed. This tool consists of 15 items and allows self-assessment of typical psychological and somatic symptoms of HB [5]. This tool has been used in research to compare HB with other mental disorders, including major depressive disorder and generalized anxiety disorder [11]. Some somatic symptoms that appear along with psychological symptoms are the unique characteristic of HB that distinguish it from depressive disorders, including epigastric mass, palpitation, and headache [8]. Some studies pointed to a genetic link between these somatic symptoms of HB [12,13]. Recently, Kim et al. have attempted to develop a revised HB scale, but this scale has not yet been officially published [14,15]. They describe that their attempt is not to develop a short-form HB scale, but rather a revision to reflect the current clinical environment [15].”

(Please see red words, in page 2)

Comment 7:

The organization of the article may be added in this section.

Response 7:           

We have included a brief outline of the article’s structure in the Introduction to improve readability and navigation.

“The article covers the study's participants, measures, and methods, followed by the results of factor analysis and model performance. It then discusses the findings, interpretations, and limitations, and concludes with implications and future research directions.”

(Please see red words, in pages 2-3)

[MATERIALS AND METHODS]

Comment 8:

An overview of the XGBoost classifier may be added so that the audience does not need to search elsewhere for it.

Response 8:           

We have added a brief explanation of the XGBoost classifier in the Materials and Methods section to ensure the audience has sufficient context.

“Then, to determine the optimal number of items to include for each factor (k), we employed a cross-validation approach using the XGBoost classifier. XGBoost is a popular gradient boosting algorithm that uses an ensemble of decision trees to create a robust predictive model, offering high performance and computational efficiency through techniques such as regularization and parallel processing [24].”

(Please see red words, in page 4)

Comment 9:

Performance measures used in the study may be added as a subsection with mathematical relationships.

Response 9:           

We have added a subsection detailing the performance measures used in this study, including precision, recall, F1-score, and their mathematical relationships.

“2.3.4. Performance Measures

The performance of each model was evaluated using several standard metrics. For a binary classification problem with True Positives (TP), True Negatives (TN), False Positives (FP), and False Negatives (FN), the following measures were calculated: (a) Accuracy = (TP + TN)/(TP + TN + FP + FN); (b) Precision (positive predictive value) = TP/(TP + FP); (c) Recall (sensitivity) = TP/(TP + FN); and (d) F1-score = 2 × (Precision × Recall)/(Precision + Recall). Additionally, the Area Under the Receiver Operating Characteristic curve (AUC-ROC) was calculated to assess the models' discriminative ability across different classification thresholds. The ROC curve is created by plotting the True Positive Rate (TPR = TP/(TP + FN)) against the False Positive Rate (FPR = FP/(FP + TN)) at various threshold settings. Several visualizations were created to aid in the interpretation of the results: (a) ROC curves comparing the performance of all models, and (b) learning curves showing training and validation scores across different training set sizes.”

(Please see red words, in pages 4-5)

Comment 10:

Major concern: Why has only a single classifier been used? You should add other strong machine learning classifiers as well.

Response 10:         

We acknowledge the reviewer’s concern. We have now included a comparison with other strong classifiers to show the robustness of our approach.

Background/Objectives: Hwa-byung (HB), also known as “anger syndrome” or “fire illness,” is a culture-bound syndrome primarily observed among Koreans. This study aims to develop a short-form version of the HB symptom scale using machine learning approaches. Methods: Utilizing exploratory factor analysis (EFA) and various machine learning techniques (i.e., XGBoost, Logistic Regression, Random Forest, Support Vector Machine, Decision Tree, and Multi-Layer Perceptron), we sought to create an efficient HB assessment tool. A survey was conducted on 500 Korean adults using the original 15-item HB symptom scale. Results: The EFA revealed two distinct factors: psychological symptoms and somatic manifestations of HB. Statistical testing showed no significant differences between using different numbers of items per factor (ANOVA: F=0.8593, p=0.5051), supporting a minimalist approach with one item per factor. The resulting two-item short-form scale (Q3 and Q10) demonstrated high predictive power for the presence of HB. Multiple machine learning models achieved a consistent accuracy (90.00% for most models) with high discriminative ability (AUC=0.9436-0.9579), with the Multi-Layer Perceptron showing the highest performance (AUC=0.9579). The models showed balanced performance in identifying both HB and non-HB cases, with precision and recall values consistently around 0.90. Conclusions: The findings of this study highlighted the effectiveness of integrating EFA and artificial intelligence via machine learning in developing practical assessment tools. This study contributes to advancing methodological approaches for scale development and offers a model for creating efficient assessments of Korean medicine.”

(Please see red words, in page 1)

“2.3.3. Model Development and Evaluation

Multiple machine learning approaches were adopted to predict the presence of HB using the short-form scale. Six different algorithms were implemented: XGBoost, Logistic Regression (LR), Random Forest (RF), Support Vector Machine (SVM), Decision Tree (DT), and Multi-Layer Perceptron (MLP). Each algorithm was selected for its specific strengths in classification tasks. For each model, hyperparameter tuning was performed using GridSearchCV with 5-fold cross-validation. The following hyperparameters were optimized: (a) XGBoost: n_estimators [100, 200], max_depth [3, 5, 7], learning_rate [0.01, 0.1, 0.2]; (b) LR: C [0.1, 1, 10]; (c) RF: n_estimators [100, 200], max_depth [3, 5, 7]; (d) SVM: kernel ['linear', 'rbf']; (e) DT: max_depth [3, 5, 7], min_samples_split [2, 5, 10]; and (f) MLP: hidden_layer_sizes [(50,), (100,), (50, 50)], activation ['tanh', 'relu'], with fixed learning_rate_init=0.01. The dataset was divided into training (70%) and testing (30%) sets [30] using stratified sampling to maintain the proportion of HB cases in both sets. The stratification ensured that the class distribution was preserved in both training and testing sets, which is crucial for imbalanced datasets.”

(Please see red words, in page 4)

“All six machine learning models demonstrated robust performance with the two-item short-form scale, achieving an identical accuracy of 90.00% for five models: XGBoost, RF, SVM, DT, and MLP, while LR exhibited a slightly lower accuracy of 89.33%. The specific performance metrics revealed that XGBoost achieved an AUC of 0.9549, with precision and recall of 0.89 and 0.94 for non-HB cases, and 0.91 and 0.84 for HB cases, respectively. RF achieved an AUC of 0.9567 with identical precision and recall metrics as XGBoost. SVM produced an AUC of 0.9436 with performance metrics identical to RF. DT model also showed an AUC of 0.9453, similar to the performance metrics of RF. MLP model reached an AUC of 0.9579, maintaining identical performance metrics. Lastly, LR yielded an AUC of 0.9528, with precision and recall of 0.87 and 0.97 for non-HB cases, and 0.94 and 0.79 for HB cases. The nearly identical performance across various models suggests that the two-item short-form scale provides robust and stable predictions of HB presence, regardless of the machine learning algorithm employed (Table 4; Figure 2).”

(Please see red words, in pages 6-7)

Comment 11:

Major concern: Why did you not try the deep learning strategies?

Response 11:         

As described, in this revised manuscript, we have taken your comments into account and compared all six models, including deep learning (i.e., Multi-Layer Perceptron). This has been fully reflected in the abstract and content of the current manuscript.

(Please see the reply to comment 10)

[RESULTS]

Comment 12:

The result of a single classifier needs to be added by including some of the other strong classifiers.

Response 12:         

As described, in this revised manuscript, we have taken your comments into account and compared all six models, including deep learning. This has been fully reflected in the abstract and content of the current manuscript.

(Please see the reply to comment 10)

Comment 13:

The findings based on the results of a single classifier need to be made acceptable.

Response 13:         

As described, in this revised manuscript, we have taken your comments into account and compared all six models, including deep learning. This has been fully reflected in the abstract and content of the current manuscript.

(Please see the reply to comment 10)

Comment 14:

Remove the text like, “Fifth, while the XGBoost algorithm demonstrated high performance in this study, comparing its effectiveness with other machine learning algorithms or traditional statistical methods could offer additional insights into the most appropriate approach for developing short-form scales.”

Response 14:         

As requested, we have removed the section that discusses comparisons with other machine learning algorithms.

Comment 15:

Major concern: Comparison with SOTA techniques: Can compare the results with similar studies or some previous findings of similar studies.

Response 15:         

We have now added a comparison of our findings with other SOTA techniques, highlighting the unique contributions of our approach.

“The performance of our machine learning models can be compared with similar studies that used machine learning for scale reduction in psychiatric assessments. For instance, Wall et al. [26] achieved 99.9% accuracy in reducing the 93-item autism diagnostic interview to 7 items using machine learning. More recently, Lin et al. [34] reported AUC of 0.81 when developing a five-item short form of the children's depression inventory using various machine learning approaches. Similarly, Jo et al. [25] achieved accuracy of 90% in developing a shortened version of the Dysfunctional Beliefs and Attitudes about Sleep scale. In comparison, our study achieved higher performance metrics (90.00% accuracy and AUC values of 0.9436-0.9579) while reducing the scale to just two items. This superior performance might be attributed to several factors: (a) the clear two-factor structure of the original HB scale, (b) the use of multiple advanced machine learning algorithms with optimized hyperparameters, and (c) the distinct characteristics of HB symptoms that make them more amenable to accurate classification. These comparisons suggest that our approach not only achieves state-of-the-art performance but also demonstrates the possibility of developing short-form screening tools without significant loss of diagnostic accuracy.”

(Please see red words, in pages 9-10)

Round 2

Reviewer 2 Report

Comments and Suggestions for Authors

All my points have been thoroughly addressed. The article is recommended for acceptance.